# Detection of Germline Mutations in a Cohort of 250 Relatives of Mutation Carriers in Multigene Panel: Impact of Pathogenic Variants in Other Genes beyond *BRCA1/2*

**DOI:** 10.3390/cancers15245730

**Published:** 2023-12-06

**Authors:** Sara Di Rado, Roberta Giansante, Michela Cicirelli, Lucrezia Pilenzi, Anastasia Dell’Elice, Federico Anaclerio, Martina Rimoldi, Antonino Grassadonia, Simona Grossi, Nicole Canale, Patrizia Ballerini, Liborio Stuppia, Ivana Antonucci

**Affiliations:** 1Center for Advanced Studies and Technology (CAST), “G. D’Annunzio” University of Chieti-Pescara, 66100 Chieti, Italy; sara.dirado@phd.unich.it (S.D.R.); roberta.giansante@phd.unich.it (R.G.); michela.cicirelli@studenti.unich.it (M.C.); lucrezia.pilenzi@studenti.unich.it (L.P.); federico.anaclerio@unich.it (F.A.); antonino.grassadonia@unich.it (A.G.); patrizia.ballerini@unich.it (P.B.); stuppia@unich.it (L.S.); i.antonucci@unich.it (I.A.); 2Department of Medical Genetics, “G. D’Annunzio” University of Chieti-Pescara, 66100 Chieti, Italy; 3SD Genetica Medica, IRCCS Fondazione Ca’Granda Ospedale Maggiore Policlinico, 20122 Milano, Italy; martina.rimoldi@policlinico.mi.it; 4Department of Innovative Technologies in Medicine and Dentistry, “G. D’Annunzio” University of Chieti-Pescara, 66100 Chieti, Italy; 5U.O.C. Chirurgia Generale ad Indirizzo Senologico, Eusoma Breast Center ASL2 Abruzzo, 66026 Ortona, Italy; sgrossi@unich.it (S.G.); canalenicole@gmail.com (N.C.)

**Keywords:** NGS multigene panel, hereditary cancer, healthy collateral family members

## Abstract

**Simple Summary:**

During the last few decades, the basis for a genetic predisposition for several cancer syndromes has been clarified, and the highly penetrant/high-risk genes mutated in familial cases are currently subjected to genetic diagnostic screening programs. Mutation testing in these genes has a major impact on genetic counseling, defines the prognosis of carriers, identifies the most appropriate and personalized prophylactic measures, and increases the chance of survival. We aim to underline the effectiveness of the multigene panel in increasing the detection rate of germline mutations in cancer patients and consequently improve the healthy carriers’ identification.

**Abstract:**

Background: Several hereditary–familial syndromes associated with various types of tumors have been identified to date, evidencing that hereditary cancers caused by germline mutations account for 5–10% of all tumors. Advances in genetic technology and the implementation of Next-Generation Sequencing (NGS) have accelerated the discovery of several susceptibility cancer genes, allowing for the detection of cancer-predisposing mutations in a larger number of cases. The aim of this study is to highlight how the application of an NGS-multigene panel to a group of oncological patients subsequently leads to improvement in the identification of carriers of healthy pathogenic variants/likely pathogenic variants (PVs/LPVs) and prevention of the disease in these cases. Methods: Starting from a total of 110 cancer patients carrying PVs/LPVs in genes involved in cancer susceptibility detected via a customized NGS panel of 27 cancer-associated genes, we enrolled 250 healthy collateral family members from January 2020 to July 2022. The specific PVs/LPVs identified in each proband were tested in healthy collateral family members via Sanger sequencing. Results: A total of 131 out of the 250 cases (52%) were not carriers of the mutation detected in the affected relative, while 119 were carriers. Of these, 81/250 patients carried PVs/LPVs on *BRCA1/2* (33%), 35/250 harbored PVs/LPVs on other genes beyond *BRCA1* and *BRCA2* (14%), and 3/250 (1%) were PVs/LPVs carriers both on *BRCA1/2* and on another susceptibility gene. Conclusion: Our results show that the analysis of *BRCA1/2* genes would have only resulted in a missed diagnosis in a number of cases and in the lack of prevention of the disease in a considerable percentage of healthy carriers with a genetic mutation (14%).

## 1. Introduction

Several hereditary–familial syndromes associated with various types of tumors have been identified to date. The most common are Lynch syndrome (hereditary non-polyposis colorectal cancer, HNPCC) and breast and ovarian cancer syndrome (HBOC). However, there are many other syndromes, such as familial adenomatous polyposis (FAP), Cowden syndrome, and Li Fraumeni syndrome [1,2], related to germline mutations in genes less frequently involved in hereditary cancer but that can be transmitted via inheritance, increasing the risk of cancer within family members. Consequently, the prevalence of hereditary tumors, considered to account for 5–10 percent of all cancers [3], could likely be underestimated.

During the last few decades, the basis for such genetic predisposition has been clarified for several hereditary cancer syndromes, and highly penetrant/high-risk genes mutated in familial cases are currently subjected to genetic diagnostic screening programs [4]. Mutation testing in these genes has a major impact on genetic counseling, defines the prognosis of carriers, identifies the most appropriate and personalized prophylactic measures, and increases the chance of survival. In HBOC, the highly penetrant *BRCA1* and *BRCA2* susceptibility genes were discovered between 1994 and 1995 [5]. Subsequent genetic studies based on linkage and positional cloning helped identify additional moderate-risk genes, and genome-wide association studies identified common low-penetrance alleles associated with breast cancer heritability [5]. In Lynch syndrome, germline pathogenic variants in the mismatch repair (MMR) genes *MLH1*, *MSH2*, *MSH6*, and *PMS2* play an essential role in carcinogenesis. Importantly, these genes have variable penetrance and different risk rates of endometrial and colon cancer; in particular, *MSH6* and *PMS2* are estimated to have lower penetrance for colorectal cancer [6]. In this context, multigene panel testing is considered a powerful tool for increasing the detection rate of pathogenic variants in a number of non-BRCA genes and should be routinely supplied to high-risk patients. As a result, the use of NGS technology in clinical practice is expanding.

The majority of hereditary–familial syndromes are inherited in an autosomal dominant manner. Once a causative mutation has been identified in a patient, there is an indication to extend the analysis to first-degree relatives. In fact, each family member of an individual carrying the mutation has a 50% chance of being a carrier of the same mutation. In a few cases, however, mutations in both alleles are required to produce a high oncological risk, deemed autosomal recessive inheritance. In these latter cases, genetic testing is first indicated for the siblings of the index case because each of them has a 25% chance of having inherited both mutations [7].

It is critical to emphasize that if a family member does not inherit the pathogenetic mutation, their risk is similar to that of the general population. In contrast, in the presence of the causative mutation, the risk of developing the disease during a lifetime is higher [8].

According to national and international guidelines, surveillance protocols for healthy mutation carriers include imaging and laboratory tests, depending on the genetic mutation detected. For female *BRCA* mutation carriers, instrumental surveillance for breast and ovarian cancer is suggested, while for male *BRCA* carriers, surveillance for breast and prostate cancer is planned. Screening protocols allow for early diagnosis and prompt treatment in order to have a better prognosis [9].

In the last year, an increasing number of studies have suggested the use of multigene panel analysis including low-, moderate-, and high-penetrance genes [10]; a crucial point of the present study is to evaluate how this evolution in the detection of cancer-predisposing mutations can affect our ability to identify healthy carriers and prevent the disease in these subjects.

The purpose of this manuscript is to underline the effectiveness of the multigene panel in increasing the detection rate of germline mutations in cancer patients and, as a result, in improving the identification of healthy collateral family members. We first collected 110 cancer patients who were carriers of PVs/LPVs detected using a customized NGS panel of 27 cancer-associated genes, and subsequently, we proceeded with the detection of known mutations in healthy collateral family members.

## 2. Materials and Methods

### 2.1. Study Population

A retrospective study was carried out on 250 subjects (155 women and 95 men) who were relatives of 110 cancer patients carrying a PV/LPV in the *BRCA1/2* genes or other cancer susceptibility genes and referred to the Medical Genetic Service of the University “G.d’Annunzio” of Chieti-Pescara–Center of Advanced Studies and Technologies (CAST) from January 2020 to July 2022. Among them, 44 cancer patients entered the study belonging to families in which the mutation was detected in another affected relative. All cases’ medical personal and family histories were acquired during genetic counseling in the presence of a clinical multidisciplinary team based on geneticists and psychologists. All patients were informed about the significance of the genetic test and the possible implications of detecting the gene variant related to an increased cancer risk and possible prevention strategies. All subjects signed an informed consent form. The results obtained from the analysis and their implications were explained during the post-test counseling.

### 2.2. Genomic DNA Extraction

Buccal swabs or blood samples were collected from all patients. Genomic DNA was extracted using the MagPurix instrument and the Forensic DNA Extraction Kit (Zinexts Life Science Corp, Taipei, Taiwan-CatZP01001)/Blood DNA Extraction Kit 200 (Zinexts Life Science Corp, Taipei, Taiwan-CatZP02001), according to the manufacturer’s protocol.

### 2.3. Next-Generation Sequencing (NGS)

NGS analysis was carried out with a Thermo-fisher Oncomine custom panel developed in our laboratory, including 27 genes (Table 1). NGS was performed via the Ion Torrent S5 system (Thermo Fisher Scientific, Waltham, MA, USA) after automatic library preparation using Ion Chef (Thermo Fisher Scientific, Waltham, MA, USA). Ion Chef consists of fragmentation and adapter ligation onto the PCR products, called clonal amplification. After quantification of DNA libraries with the Real-Time Step One PCR System (Thermo Fisher Scientific, Waltham, MA, USA), the prepared samples of ion sphere particles (ISP) were loaded onto an Ion 530™ chip with the Ion Chef (Thermo Fisher Scientific, Waltham, MA, USA). Sequencing was performed using the Ion S5™ sequencing reagents (Thermo Fisher Scientific, Waltham, MA, USA). The Torrent Suite 5.14.0 platform and specific plugins were used for NGS data analysis. The uniformity of base coverage was over 98% in all batches, and base coverage was over 20× in all target regions.

### 2.4. Sanger Sequencing

The specific PVs/LPVs identified in each proband via NGS were tested in healthy collateral family members enrolled in the study via Sanger sequencing. All DNA samples were amplified via polymerase chain reaction (PCR) performed in 30 μL reaction volume, containing 22.25 μL of H_2_O, 3 μL of 10X PCR buffer, 2.1 μL of MgCl2 solution 25 mM, 0.5 μL of dNTPs 10 mM, 0.15 μL of AmpliTaq Gold polymerase, 1 μL of DNA, and 0.5 μL of Forward and 0.5 μL of Reverse primers. All primers were designed using NCBI designing tools (https://www.ncbi.nlm.nih.gov/tools/primer-blast/ accessed on 12 October 2023).

Amplification was performed via SimpliAmpTM thermal cycler (ThermoFisher, Applied Biosystem, CA, USA). FastGene Gel/PCR Extraction (Nippon Genetics Europe, Düren, Germany) was utilized for the purification of the PCR products, according to the manufacturer’s protocol. The amplification products were submitted to direct sequencing procedure using BigDye Term v3.1 CycleSeq Kit (Life Technologies, Monza, Italy) followed by automatic sequencing analysis. All sequences were purified via “NucleoSEQColumns” purification kit (Macherey-Nagel Colonia, Dueren, Germany) and analyzed in forward and reverse directions on a SeqstudioGenetic Analyzer (ThermoFisher, Applied Biosystem, Foster City, CA, USA).

### 2.5. Genetic Variant Classification

According to the guidelines of the Evidence-based Network for the Interpretation of Germline Mutant Alleles (ENIGMA) (https://enigmaconsortium.org/ accessed on 12 October 2023), the genetic variants were classified into five classes: benign (C1), likely benign (C2), variant of uncertain significance (VUS, C3), likely pathogenic (C4), and pathogenic (C5). In our study, we focused on the LPVs and PVs that can be used for clinical purposes and cancer prevention. The variants were referred to according to the nomenclature recommendations of the Human Genome Variation Society (https://www.hgvs.org accessed on 12 October 2023). The clinical significance of the genetic variants found in this study was evaluated according to ClinVar (https://www.ncbi.nlm.nih.gov/clinvar/ accessed on 12 October 2023), Varsome (https://varsome.com accessed on 12 October 2023), Franklin Genoox (https://franklin.genoox.com accessed on 12 October 2023) and, for some other susceptibility genes, according to LOVD-InSIGHT (https://www.insight-group.org/variants/databases/ accessed on 12 October 2023).

## 3. Results

Starting from 110 affected probands, tested with a NGS multigene panel based on 27 cancer susceptibility genes found in a total of 250 healthy relatives analyzed via Sanger sequencing, we detected 119 cases harboring at least one PVs/LPVs. Importantly, 26 out of these 119 relatives tested had cancer and pathogenic variants (Table 2).

A total of 143 were aged <45 years old and 107 were aged >45 years old. One hundred nineteen cases were detected to be carriers of the mutation previously evidenced in an affected relative. Of these, 81 cases had PVs/LPVs on BRCA1/2 (33%), 35 in other genes related to cancer susceptibility (14%), and only 3 patients had PVs/LPVs on both BRCA1/2 and other genes (1%) (Figure 1). One hundred thirty-one patients did not inherit the pathogenic mutation previously detected in the family (52%). Among the younger group, 53 had BRCA1/2 germline PVs/LPVs (38%), 15 were carriers of other cancer susceptibility genes (10%), primarily APC, NBN, ATM, MUTYH, MLH1, and only 2 patients were carriers of PVs/LPVs in both BRCA1/2 and other susceptibility genes (1%) (Figure 2). In total, 29 out of the 53 BRCA1/2 PVs/LPVs carriers were female and 24 were male; meanwhile, among the 15 carriers of other susceptibility genes, 6 were female and 9 were male. Seventy-three patients showed no PVs/LPVs (51%).

In the older group, 28 were carriers of BRCA1/2 germline PVs/LPVs (26%), while 20 had mutations on other genes (18%), such as CHEK2, MUTYH, PALB2 and BRIP1. Only one patient carried a PV/LPV in both BRCA2 and ATM (Figure 2). Fifty-eight cases had no PVs/LPVs (55%). Among the 28 BRCA1/2 carriers, 19 were female and 9 were male; meanwhile, in the other susceptibility genes carriers’ group (20 patients), 9 were female and 11 were male.

Overall, the most prevalent PV/LPV on BRCA1 was c.5266dupC, while on BRCA2, it was c.7007G>A; these were found, respectively, in seven and four patients from different families.

Specifically, the BRCA1 variant causes an insertion of one cytosine, resulting in a frameshift mutation with the creation of a novel translational termination codon after 74 amino acid residues [p.(Gln1756Profs*74)]. The protein product thus produced is truncated and non-functional [11].

The BRCA2 pathogenic variant, instead, replaces arginine with histidine at codon 2336 of the protein [p.(Arg2336His)]. RNA analysis indicates that this missense mutation induces altered splicing and may result in an absent or disrupted protein product [12]. Another interesting finding was the presence of germline PVs/LPVs on BRCA2 in 22 out of 45 male carriers (49%).

Our analysis revealed that CHEK2 was the gene with the most recurrent mutations, found in 11 patients, while the second most mutated gene was MUTYH, found in 5 patients. Two out of five patients (40%) with MUTYH mutation showed the c.884C>T p.(Pro267Leu) variant.

The most frequent CHEK2 PV/LPV was c.499G>A, observed in nine individuals (9/11, 82%); this missense variant located in coding exon 3 of the gene results from a Guanine to Adenine substitution at nucleotide position 499 and has a deleterious impact on protein structure and function [p.(Gly167Arg)] [13]. In Figure 3, a family segregation study in which eight patients aged >45 years were found to carry the same proband PV c.499G>A in the CHEK2 gene is presented. The proband is an 85-year-old woman diagnosed with breast cancer at age 54 and then at age 68, and diagnosed with endometrial cancer at age 83. The proband’s children, aged 51 and 53, are both mutated but currently healthy. The proband has five siblings, including one who is a carrier of the variant in CHEK2 and has had chronic myeloid leukemia since age 65, three siblings who are carriers of the variant and are currently healthy collaterals, and only one who has not inherited the variant. Moreover, the proband also has two sisters, both carriers of the variant, but only one received a breast cancer diagnosis at age 67 years. Furthermore, in another family with 10 healthy collateral relatives of a proband with pancreatic cancer at the age of 58 years, we tested two variants: BRCA2 c.8487+1G>A and ATM c.6095G>A. Seven patients were carriers of the c.8487+1G>A pathogenic variant in BRCA2, and three were carriers of both variants (Figure 4). One relative is the proband’s sister with breast cancer at age 46. From the segregation analysis, both children, a 29-year-old male and a 26-year-old female, inherited both mutations, although they are currently still healthy carriers. In addition, both of the proband’s sons, one 39 years old and the other 33 years old, inherited only one of the two mutations and are currently healthy collaterals. Another sister of the proband and her son are both mutated in BRCA2, but are currently healthy collaterals. These hereditary cases showed a relevant penetrance of variants in other genes beyond BRCA1/2.

Analyzing the healthy collateral family members, one family showed strong inheritance with a PV on BRCA2, the c.6450dupA one, found in all of the five family members tested. Specifically, three were early onset and two were late onset.

In particular, the BRCA2 intronic variant occurs in the invariant region of the splice consensus sequence and is predicted to cause altered splicing leading to an abnormal or absent protein [14]; the missense variant in ATM causes a G to A nucleotide substitution at the last nucleotide of exon 41 of the ATM gene and replaces arginine with lysine at codon 2032 of the ATM protein [p.(Arg2032Lys)]. The aberrant transcript is expected to result in an absent or non-functional protein product [15].

## 4. Discussion

Hereditary tumors caused by germline mutations account for 5–10% of all cancers, with increased prevalence in some specific cancers such as breast, ovary, colon, and others. Advances in genetic technology and the implementation of Next-Generation Sequencing (NGS) have accelerated the simultaneous analysis of several susceptibility cancer genes. In fact, even though the *BRCA1/2* genes are known to explain up to 25% of all the suspected hereditary forms [16,17], several other non-BRCA genes are known to be involved in cancer predisposition, as evidenced by the continuous updating of the National Comprehensive Cancer Network’s (NCCN) guidelines for hereditary cancers [18]. As a consequence of this improvement in the diagnosis of hereditary cancers, a larger number of cancer patients are at present identified as carriers of genetic mutations, increasing their risk of developing cancer during their lifetime [19,20]. In turn, this leads to an increased number of healthy relatives in which the presence of the mutation must be assessed to prevent the disease’s development. The aim of the present study is to highlight how the application of the multigene panel on cancer probands can subsequently improve the healthy non-BRCA PVs/LPVs carriers’ identification. There is no literature data on cascade genetic testing for genes with low and moderate penetrance. The present study, for the first time, investigated the role of cascade testing in at-risk relatives of a proband with PVs/LPVs in non-BRCA genes. Several clinical guidelines recommend the use of this strategy; however, the majority of research has focused on families with hereditary breast and ovarian cancer, as well as Lynch syndrome [21,22,23]. Our study suggests cascade testing should be implemented and some barriers should be overcome: intra-familial communication rates, depression or anxiety of probands or relatives, and limited surveillance protocols for individuals with a mutation in moderate- and low-penetrance genes. Unfortunately, there are no national guidelines or recommendations in Italy, and as a result, access to cascade genetic testing varies by region and even within the same region. Considering these difficulties, some tangible proposals could be put forward: (a) appropriate genetic counselling; (b) clear clinical guidelines for individuals with a mutation in moderate- and low-penetrance genes; (c) accurate risk assessment; (d) improved rates of communication between genetics professionals and probands in order to encourage them to discuss results with their families; and (e) following post-test counseling with the patients, it would be appropriate to write a family letter to communicate the test results and discuss their implications with relatives. It is important to emphasize that a crucial point in applying information about the gene variant in cancer prevention is related to the different risks associated with each single gene. In other words, the prevention strategies to use in patients with non-*BRCA1/2* PVs/LPVs are different from those typically adopted in *BRCA1/2* carriers. The NCCN Clinical Practice Guidelines in Oncology have specific recommendations for patients found to have pathogenic variants that confer an increased risk of breast cancer, including imaging modalities, frequency of evaluation and risk-reducing surgery. Genetic testing and NCCN guidelines for patients with pathogenic variants have changed the clinical landscape for breast oncologists, who routinely address the relevance of genetics, the criteria for testing, and recommendations for radiographic and/or operative follow-up during patient consultations [18,24].

Regarding moderate-risk breast cancer susceptibility genes, *CHEK2* was the most frequently mutated gene in our population. *CHEK2* is a tumor suppressor gene conferring a predisposition to sarcoma, breast cancer, and brain tumors. *CHEK2*, a protein kinase activated in response to DNA damage, is involved in cell cycle arrest, and heterozygous germline mutations in this gene have been reported in patients with Li-Fraumeni syndrome-2 [25]. Several studies have demonstrated that pathogenic mutations in *CHEK2* occur at a higher frequency than those reported in other genes in multigene panels [26,27,28], which is consistent with our findings. The relationship between *CHEK2* PVs and the prognosis of BC is still unknown. One case example from our cohort was a woman who received a diagnosis of bilateral breast cancer at age 54 and 68 and a diagnosis of endometrial cancer at age 83. Based on limited data, the evidence does not support the relationship of *CHEK2* mutations with a significantly higher risk of endometrial cancer [29]. As a result, healthy carriers of *CHEK2* pathogenic mutations were offered breast and colorectal cancer surveillance in accordance with NCCN and AIOM (Italian Association of Medical Oncology) while also taking their family history into account. Specifically, the suggested prevention protocol provides, for female patients, breast clinical and instrumental surveillance, with an annual mammography and magnetic resonance imaging (MRI) starting at the age of 40. In contrast, for men, the prevention protocol requires an annual Prostate Specific Antigen (PSA) dosage for prostate cancer surveillance, starting at the age of 40. The second most frequently mutated gene in our population is *MUTYH*, defined as a “high-penetrance” gene. Mammalian MutY homologue (*MUTYH*) encodes a DNA glycosylase involved in base excision repair during DNA replication and damage repair. PVs/LPVs in *MUTYH* are associated with autosomal recessive colorectal adenomatous polyposis, but interestingly, monoallelic variants on this gene have been reported by both our and other groups as being associated with cancer predisposition in several patients [30,31]. An interesting case in the present study is represented by the detection of a *MUTYH* c.734G>A variant in one female patient with a personal history of breast cancer diagnosed at the age of 44 years, previously tested for *BRCA1/2* variants at another institute and found to be negative. In this case, the identification of the pathogenic variant was achieved by using the multigene panel testing in her sister, suggesting the usefulness of multigene panel analysis in affected patients negative for *BRCA1/2* testing in the presence of strong familiarity. In addition, another female patient was detected to harbor only the c.650G>A in the *MUTYH* gene while being negative for the second variant (c.884C>T) found in the proband (a son affected by colon cancer). The patient had a personal history of cancer; first, she had a diagnosis of breast cancer at the age of 50, and then of colon cancer at an older age (82 years). Due to the time of disease onset, she had never received the indication for genetic testing, representing a further case of detection of a germline mutation through the analysis of an affected relative using a multigene panel. To date, the cancer risk associated with germline variants in individuals carrying only one *MUTYH* defective allele is controversial. Studies have shown that the risks of colorectal cancer for carriers of monoallelic variants in *MUTYH* with a first-degree relative with colorectal cancer are sufficiently high to warrant more intensive screening than for the general population; as a consequence, NCCN guidelines propose a colonoscopy every five years beginning at age 40 [32,33]. Nevertheless, there is no strong evidence of an association between increased BC risk and carriers of monoallelic variants in *MUTYH* [34]. More research is needed to confirm the cancer risks linked to these heterozygous *MUTYH* mutations. Some considerations should also be made about the *ATM* variants encountered in our cohort. Focusing on family B (see Figure 4), the proband who initiated the segregation analysis was diagnosed with pancreatic tail and body cancer at the age of 55 and was found to carry PVs/LPVs in two distinct genes, specifically the c.6095G>A in *ATM* and the c.8487+1G>A in *BRCA2*. Sanger sequencing in healthy collaterals highlighted the presence of the same compound heterozygosity in the proband’s daughter and in his two male grandchildren, with a negative personal oncological anamnesis. To date, there are few published cases of double heterozygosity (DH) in cancer-predisposing genes and there is still no clear evidence on the incidence of such digenic abnormalities and the severity of illness symptoms [35,36]. Although DH pathogenic variants in *BRCA2* and *ATM* have been previously reported in the literature, the functionality of the gene product and clinical details should be better elucidated [37]. Women carriers of pathogenic variants in the *BRCA2* gene have an increased risk of developing early-onset BC and PC. While heterozygous carriers with *ATM* gene mutations are more likely to develop BC, the risk for other neoplasms remains controversial. In comparison to the high-penetrance *BRCA1/2* genes, pathogenic mutations of the *ATM* gene have a more modest penetrance. In view of these considerations, the coexistence of two PVs may have contributed to our patient’s diagnosis of early PC. It is also crucial to note that the severity of the illness phenotype in double versus single heterozygotes is debatable. Several significant clinical questions regarding the management of double heterozygotes remain unanswered, including whether the higher-penetrance gene should dictate the cancer surveillance schedule. However, we propose in our center that DH carriers have more thorough surveillance and follow-up care, and that family members conduct cascade testing. Specifically, according to the NCCN guidelines, the suggested prevention protocol includes the following: an annual mammography and magnetic resonance imaging (MRI) starting from the age of 40 for breast cancer; a clinical instrumental surveillance regarding annual gynecologic examination with transvaginal ultrasound from age 40 years and an annual CA-125 assay from the age of 40 years for ovarian cancer; gastroscopy and endoscopic ultrasound from age 40 years for gastric cancer; colonoscopy every 5 years starting at age 40 years for colorectal cancer and for pancreatic cancer ultrasound; and possibly MRI if there have been relatives with pancreatic cancer in the family, starting at age 45 years or 10 years earlier if there have been cases of juvenile pancreatic cancer. In future, functional studies should be carried out to clarify how rare DH genotypes accelerate and impact carcinogenesis. In particular, further investigations are needed to better understand if the simultaneous presence of two haploinsufficient genes is more harmful than the presence of a single-gene deficiency. Finally, special emphasis is placed on the role of modifier genes and environmental factors in preventing cancer-related morbidity and mortality and current research is attempting to solve this puzzle.

## 5. Conclusions

In conclusion, from the analysis of our data, it emerges that without the application of the NGS multigene panel in the probands, a considerable percentage of healthy collaterals who are carriers of PVs/LPVs in other susceptibility genes would have been lost (14%). This percentage corresponds to 35 healthy carriers who, due to the presence of germline variants, will be included in the clinical and instrumental surveillance protocols.

The identification of hereditary forms, related to germline, inherited DNA variants, is therefore crucial to admit patients and their at-risk family members to the most proper surveillance and therapeutic programs [38]. Importantly, for the collaterals tested who presented cancers but no PVs/LPs, the first hypothesis is that it is a sporadic tumor since these account for 90% of all cancers. Finally, in the future, impacted collaterals not harboring the found family pathogenic variant could be perfect candidates for an extended molecular analysis to detect additional susceptibility genes and potential target therapeutics for better clinical illness management.

## Figures and Tables

**Figure 1 cancers-15-05730-f001:**
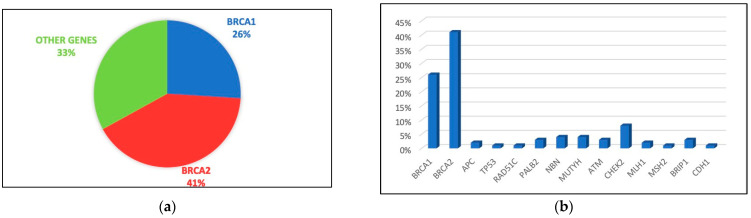
This figure shows the percentage of PVs/LPVs in 27 genes of NGS multigene panel: (**a**) shows the percentage of total PVs/LPVs found in BRCA1/2 and other susceptibility genes; (**b**) shows the percentage of PVs/LPVs for each individual gene.

**Figure 2 cancers-15-05730-f002:**
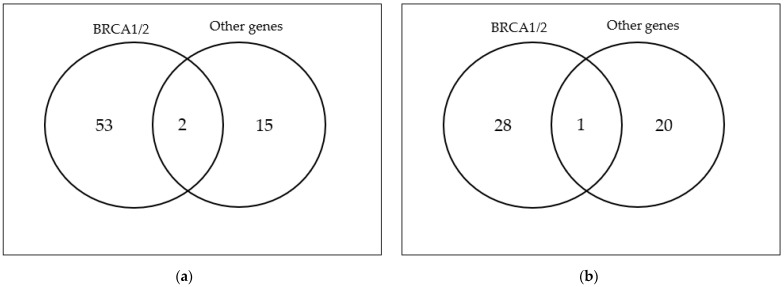
Venn diagrams: (**a**) A total of 53 carriers mutated in BRCA1/2, 15 mutated in other genes, and 2 mutated on both BRCA1/2 and other genes under 45 years; (**b**) A total of 28 carriers mutated in BRCA1/2, 20 mutated in other genes and 1 mutated on both BRCA1/2 and other genes over 45 years.

**Figure 3 cancers-15-05730-f003:**
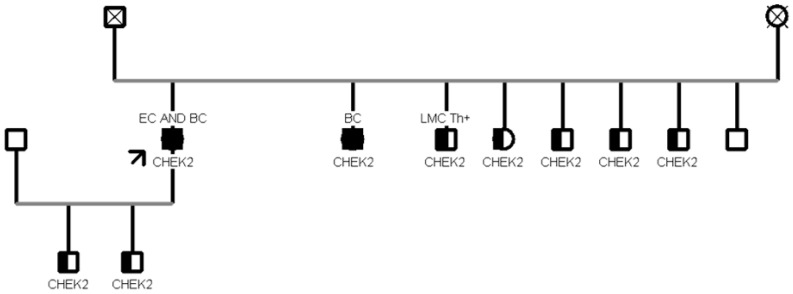
Family tree A shows the c.499G>A variant of the CHEK2 gene in eight healthy collateral relatives of the proband (pointed by the arrow). CHEK2 is the gene with the most recurrent mutations and c.499G>A is the most frequent CHEK2 PV/LPV.

**Figure 4 cancers-15-05730-f004:**
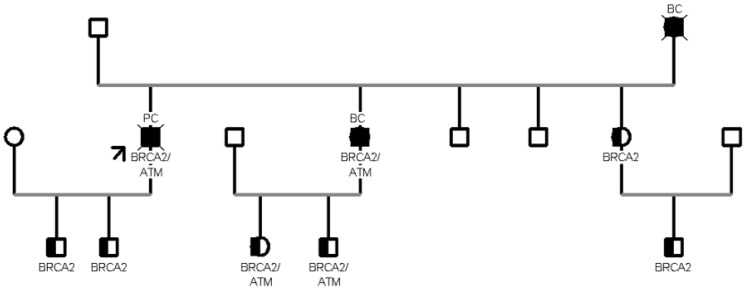
In family tree B, two different variants in BRCA2 and ATM genes are presented. Seven healthy collateral relatives of the proband (pointed by the arrow) are carriers of BRCA2 c.8487+1G>A and three are carriers of both variants (BRCA2 c.8487+1G>A/ATM c.6095G>A).

**Table 1 cancers-15-05730-t001:** Oncomine NGS panel containing 27 cancer susceptibility genes.

Gene	Omim	Refseq	Gene	Omim	Refseq
*ATM*	607,585	NM_000051.3	*PALB2*	610,355	NM_024675.3
*EPCAM*	185,535	NM_002354.2	*MLH1*	120,436	NM_000249.3
*MSH2*	609,309	NM_000251.2	*MSH6*	600,678	NM_000179.2
*PMS2*	600,259	NM_000535.6	*RAD51C*	602,774	NM_058216.2
*BRIP1*	605,882	NM_03204.2	*RAD51D*	602,954	NM_002878.3
*TP53*	191,170	NM_000546.5	*CHEK2*	604,373	NM_007194.3
*CDH1*	192,090	NM_004360.4	*PTEN*	601,728	NM_000314.6
*MUTYH*	608,456	NM_001128425.2	*APC*	611,731	NM_000038.6
*SMAD4*	600,993	NM_005359.6	*POLE*	174,762	NM_006231.3
*POLD1*	174,761	NM_001256849.1	*CDK4*	123,829	NM_000075.3
*BARD1*	601,593	NM_000465.3	*CDKN2A*	600,160	NM_000077.5
*CDK12*	615,514	NM_016507.3	*NBN*	6,026,667	NM_002485.4
*BRCA1*	113,705	NM_007294.4	*BRCA2*	164,757	NM_000059.3
*NF1*	162,200	NM_001042492.2			

**Table 2 cancers-15-05730-t002:** Description and distribution of the PVs/LPVs in our cohort.

	PVs/LPVs	Age	Type of Tumor
Subject 1	BRCA1 c.1297delG	>45	
Subject 2	BRCA1 c.1297delG	<45	
Subject 3	BRCA1 c.1297delG	<45	
Subject 4	BRCA1 c.3477_3480delAAAG	<45	
Subject 5	BRCA2 c.8487+1G>A	<45	
Subject 6	BRCA2 c.8487+1G>A	<45	
Subject 7	BRCA2 c.8487+1G>A	<45	
Subject 8	BRCA2 c.8487+1G>A	<45	
Subject 9	ATM c.6095G>A/BRCA2 c.8487+1G>A	>45	breast cancer
Subject 10	ATM c.6095G>A/BRCA2 c.8487+1G>A	<45	
Subject 11	ATM c.6095G>A/BRCA2 c.8487+1G>A	<45	
Subject 12	BRCA1 c.1953dup	>45	
Subject 13	BRCA1 c.1953dup	>45	
Subject 14	BRCA2 c.8487+1G>A	>45	
Subject 15	BRCA2 c.8487+1G>A	<45	
Subject 16	BRCA2 c.6486_6489delACAA	<45	
Subject 17	BRCA2 c.6275_6276del	>45	prostate cancer
Subject 18	BRCA2 c.7007G>A	<45	
Subject 19	BRCA2 c.7462A>G	<45	
Subject 20	APC c.5790_5798del	<45	colon polyposis
Subject 21	CHEK2 c.499G>A	>45	
Subject 22	BRCA2 c.4914dupA	<45	breast cancer
Subject 23	BRCA2 c.4914dupA	<45	
Subject 24	BRCA2 c.1238delT	>45	breast cancer
Subject 25	BRCA1 c.3477_3480delAAAG	>45	ovarian cancer
Subject 26	BRCA2 c.7940T>C	<45	
Subject 27	BRCA1 c.1953dup	<45	
Subject 28	BRCA1 c.4117G>T	>45	breast cancer
Subject 29	BRCA1 c.4117G>T	>45	breast cancer
Subject 30	BRCA1 c.5266dupC	<45	
Subject 31	BRCA2 c.8575C>T	<45	
Subject 32	BRCA2 c.7007G>A	<45	
Subject 33	BRCA2 c.7007G>A	>45	
Subject 34	MUTYH c.734G>A	>45	breast cancer
Subject 35	MSH2 c.1120C>T	<45	
Subject 36	BRCA2 c.6450dupA	>45	breast cancer
Subject 37	BRCA2 c.6450dupA	<45	
Subject 38	BRCA2 c.6450dupA	<45	
Subject 39	BRCA2 c.6450dupA	>45	breast cancer
Subject 40	BRCA2 c.6450dupA	<45	
Subject 41	BRCA1 c.1953dupG	>45	
Subject 42	BRCA1 c.1953dupG	>45	liver cancer
Subject 43	BRCA1 c.5266dupC	>45	breast cancer
Subject 44	BRCA1 c.5266dupC	>45	colon cancer
Subject 45	BRCA1 c.5266dupC	<45	
Subject 46	BRCA1 c.181T>G	>45	
Subject 47	BRCA1 c.181T>G	<45	
Subject 48	PALB2 c.1408A>T	>45	
Subject 49	BRCA2 c.5851del	<45	
Subject 50	BRCA1 c.5266dupC	>45	
Subject 51	BRCA1 c.5266dupC	<45	
Subject 52	BRCA1 c.5266dupC	<45	
Subject 53	BRCA2 c.658delGT	>45	
Subject 54	BRCA2 c.2979G>A	<45	
Subject 55	BRCA2 c.2944A>C	<45	
Subject 56	BRCA2 c.2808_2811delACAA	<45	
Subject 57	BRCA2 c.2808_2811delACAA	<45	
Subject 58	TP53 c.880G>T	<45	
Subject 59	BRCA2 c.8632G>A	<45	breast cancer
Subject 60	MLH1 c.1852_1854del	<45	
Subject 61	BRCA1 c.4117G>T	>45	
Subject 62	BRCA1 c.4117G>T	<45	
Subject 63	BRCA1 c.4117G>T	<45	
Subject 64	RAD51C c.904+5G>tT	<45	
Subject 65	BRCA2 c.631G>A	<45	
Subject 66	BRCA2 c.631G>A	<45	
Subject 67	BRCA2 c.631G>A	<45	
Subject 68	BRCA2 c.631G>A	<45	
Subject 69	MLH1 c.1852del	<45	
Subject 70	BRCA1 c.2077delGinsATA	>45	breast cancer
Subject 71	BRCA2 c.1238delT	<45	
Subject 72	BRCA1 c.181T>G	>45	skin cancer
Subject 73	PALB2 c.1408A>G	>45	breast cancer
Subject 74	BRCA2 c.5782G>T	>45	breast cancer
Subject 75	BRCA2 c.5782G>T	>45	
Subject 76	BRCA1 c.181T>G	<45	
Subject 77	BRCA1 c.181T>G	<45	
Subject 78	NBN c.741_742dupGG	>45	
Subject 79	NBN c.741_742dupGG	<45	
Subject 80	NBN c.741_742dupGG	<45	
Subject 81	NBN c.741_742dupGG	<45	
Subject 82	NBN c.741_742dupGG	>45	liver cancer
Subject 83	BRCA2 c.68-7T>A	<45	breast cancer
Subject 84	BRCA2 c.2979G>A	>45	colon cancer
Subject 85	BRCA2 c.857C>G	<45	
Subject 86	BRCA2 c.5238dupT	>45	
Subject 87	BRCA2 c.9699T>A	<45	
Subject 88	PALB2 c.661_662delGTinsTA	<45	
Subject 89	BRCA2 c.658_659delGT	<45	
Subject 90	BRCA2 c.7007G>A	<45	
Subject 91	BRCA2 c.7007G>A	<45	
Subject 92	APC c.904C>T	<45	
Subject 93	CHEK2 c.349A>G	<45	
Subject 94	CHEK2 c.349A>G	>45	
Subject 95	BRCA1 c.843_846delCTCA	<45	
Subject 96	BRCA1 c.843_846delCTCA	<45	
Subject 97	CHEK2 c.499G>A	>45	
Subject 98	CHEK2 c.499G>A	>45	
Subject 99	CHEK2 c.499G>A	>45	breast cancer
Subject 100	CHEK2 c.499G>A	>45	
Subject 101	CHEK2 c.499G>A	>45	
Subject 102	CHEK2 c.499G>A	>45	
Subject 103	CHEK2 c.499G>A	>45	
Subject 104	BRIP1 c.2379+1G>T	>45	
Subject 105	BRIP1 c.2379+1G>T	>45	
Subject 106	BRIP1 c.2379+1G>T	<45	
Subject 107	BRCA2 c.9959_9961del	<45	
Subject 108	BRCA2 c.9959_9961del	<45	
Subject 109	BRCA2 c.1238del	>45	
Subject 110	MUTYH c.650G>A	>45	breast cancer
Subject 111	MUTYH c.884C>T	>45	
Subject 112	MUTYH c.884C>T	<45	melanoma
Subject 113	MUTYH c.650G>A	<45	breast cancer
Subject 114	BRCA2 c.1285dup	<45	breast cancer
Subject 115	BRCA2 c.1285dup	<45	
Subject 116	BRCA1 c.4117G>T	>45	
Subject 117	CDH1 c.1429G>A	>45	
Subject 118	BRCA1 c.1953dupG	<45	
Subject 119	NBN c.741_742dupGG	>45	

## Data Availability

The data presented in this study are available on request from the corresponding author. The data are not publicly available due to privacy restrictions.

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
