# Peer review of "Detection of Germline Mutations in a Cohort of 250 Relatives of Mutation Carriers in Multigene Panel: Impact of Pathogenic Variants in Other Genes beyond BRCA1/2"

_cancers, 2023, doi:10.3390/cancers15245730_

Round 1
Reviewer 1 Report
Comments and Suggestions for Authors
The authors describe the use of NGS panels to search for pathogenic or likely pathogenic variants in genes other than BRCA1/2 in a cohort of relatives of mutation carriers. Their results confirm the validity of the use of multigene panels in the identification of healthy subjects but carriers of PV/LPV mutations. In this case, such individuals could enter ad hoc surveillance programs. Although the paper is not extremely original, it can serve as an starting point especially for small hospital centres that only use the BRCA1/2 test to identify the presence of deleterious mutations in patients with cancer but also in their relatives.
Author Response
Reviewer 1:
We thank you for your comments and appreciation of our work.
Reviewer 2 Report
Comments and Suggestions for Authors
Rado et al report an NGS-based multi gene panel testing to expand identification of patients with moderate/low penetance CPGs. 81 (out of 250) cases had a germline PV/LPV. Unsurprisingly, the commonest PVs/LPVs occurred in BRCA1/2. In addition, only 14% (35 cases) of patients had a non-BRCA1/2 PVs/LPVs when 25 more genes were tested.
Major criticisms:
The authors need to provide a table containing:
1. The PV/LPVs detected
2. The age ranges for these patients with these PV/LPVs
3. The types of cancer diagnosed with these PV/LPVs
Comments on the Quality of English Language
As above
Author Response
Reviewer 2:
Q1: The authors need to provide a table containing:
- The PV/LPVs detected
- The age ranges for these patients with these PV/LPVs
- The types of cancer diagnosed with these PV/LPVs
A1: According to the criticism of the referee, in the table 2 we reported the PVs/LPVs detected in BRCA1/2 and other susceptibility genes. Subjects were divided into two groups according to mean age: <45 and >45 years. The table shows the different types of cancer diagnosed.
Reviewer 3 Report
Comments and Suggestions for Authors
Di Rado and colleagues present a paper aimed at exploring cancer susceptibility through the screening of non-BRCA genes in cancer patient relatives. This work offers one of many attempts at detecting cancer susceptibility before cancer onset, taking advantage of the increasing availability of NGS instruments and protocols; thus, its publication is pertinent in Cancers. However, the authors must improve the presentation of their findings in several aspects so they live up to the importance of their data.
Major points
The Results section comprises the text and two figures showing familial studies; only one of them is mentioned in the text, and both lack legends. So, there is ample room for improvement, especially since the text suggests a great deal of data. For instance, it would be easier to visualize the proportion of mutation carriers in each group (lines 164-182) through graphs, perhaps Venn diagrams.
Similarly, a visual representation of the mutations found in the BRCA1, BRCA2, CHEK2, MUTYH, and other genes would be helpful. Is there any pattern in these mutations?
The familial studies in Figures 1 and 2 are merely described in the text, although it is very likely that more profound conclusions can be drawn. Please deepen the analysis.
The Discussion and Conclusion sections contain data that belongs in the Results section and is either repeated or absent from it. For example, the 119 mutations found in 250 collaterals (lines 236-241), the frequency of mutations in CHEK2 (lines 246-251) and MUTYH (lines 252-257), the 15 cancer patients with no pathogenic mutations (lines 336-340), and so on. Please group all the results in the corresponding section.
Consequently, the Discussion section offers minimal actual discussion, i.e., interpretation and possible implications of the results. This section is where the manuscript most crucially falls short of the expectations set in the abstract. How do the authors explain the presence of the found mutations? What research avenues do they recommend to follow and why? Is there any concrete suggestion to improve current screening protocols?
The sheer extent of the Conclusions section speaks volumes of the manuscript’s need for better organization. The opening paragraph (lines 288-292) and maybe even the second one (lines 293-295) duly highlight one of the main results of the paper. Another one or two main findings that complete the conclusions will be evident once the results are properly organized.
Minor points
Please provide citations for the following statements:
‘the basis for such genetic predisposition has been clarified for several hereditary cancer syndromes […]’ (Lines 56-58)
‘an increasing number of studies suggested the use of multigene panel analysis including low, moderate, and high penetrance genes’ (Lines 90-91)
It is hard to believe that ‘This research received no external funding’. Perhaps the authors received no specific grant for this particular study; still, the recruitment of 250 subjects, acquisition and processing of the corresponding number of samples, NGS, and Sanger verification demand considerable resources that must have had a source.
Author Response
Reviewer 3:
Q1: The Results section comprises the text and two figures showing familial studies; only one of them is mentioned in the text, and both lack legends. So, there is ample room for improvement, especially since the text suggests a great deal of data. For instance, it would be easier to visualize the proportion of mutation carriers in each group (lines 164-182) through graphs, perhaps Venn diagrams.
A1: As suggested by the referee, we have added a brief description of Figures 3 and 4 in the results section (pag 5 lines 201-219) and the legends under the respective figures. In addition, we added 2 Venn diagrams (Figure 2) showing the percentage of mutation carriers in each group (pag 4 lines 170-183): figure 2a shows the percentages of mutated carriers in BRCA1/2 and other genes under 45 years; while, figure 2b shows the percentages of mutated carriers in BRCA1/2 and other genes over 45 years.
Q2: Similarly, a visual representation of the mutations found in the BRCA1, BRCA2, CHEK2, MUTYH, and other genes would be helpful. Is there any pattern in these mutations?
A2: According to the suggestions of referee, we have added the Figure 1 for to have a visual representation of the mutations found in the BRCA1, BRCA2, CHEK2, MUTYH, and other genes (pag 4 lines 166-170). Figure 1a shows the percentage of total PVs/LPVs found in BRCA1/2 and other susceptibility genes. Figure 1b shows the percentage of PVs/LPVs for each individual gene.
Q3: The familial studies in Figures 1 and 2 are merely described in the text, although it is very likely that more profound conclusions can be drawn. Please deepen the analysis.
A3: We thank the referee for this precious criticism. We implemented the description of family cases in the results section (pag 5 lines 201-219) emphasizing the conclusion we have drawn (pag 5 lines 219-220).
Q4: The Discussion and Conclusion sections contain data that belongs in the Results section and is either repeated or absent from it. For example, the 119 mutations found in 250 collaterals (lines 236-241), the frequency of mutations in CHEK2 (lines 246-251) and MUTYH (lines 252-257), the 15 cancer patients with no pathogenic mutations (lines 336-340), and so on. Please group all the results in the corresponding section.
A4: We grouped all the results in the corresponding section (pag 4 lines 162-165). In addition, we modified the discussion (pag 11 lines 295-304, pag 11 lines 315-320, pag 12 lines 354-361) and conclusion (pag 13 lines 374-378) so that the data were not repeated in the different sections, as suggested by the referee.
Q5: Consequently, the Discussion section offers minimal actual discussion, i.e., interpretation and possible implications of the results. This section is where the manuscript most crucially falls short of the expectations set in the abstract. How do the authors explain the presence of the found mutations? What research avenues do they recommend to follow and why? Is there any concrete suggestion to improve current screening protocols?
A5: We thank the referee for this valuable criticism. We have specified in the "Discussion" section the role of the mutations found in our cohort (pag 10 lines 306-315, pag 11-12 lines 342-361), possible research fields (pag 12 lines 361-366), and the strategies to improve screening protocols (pag 10-11 lines 279-294).
Q6: The sheer extent of the Conclusions section speaks volumes of the manuscript’s need for better organization. The opening paragraph (lines 288-292) and maybe even the second one (lines 293-295) duly highlight one of the main results of the paper. Another one or two main findings that complete the conclusions will be evident once the results are properly organized
A6: We thank the referee for this useful comment. We structured and organized results so that the key findings are evident in the "Conclusions" section.
Q7: Please provide citations for the following statements:
‘the basis for such genetic predisposition has been clarified for several hereditary cancer syndromes […]’ (Lines 56-58)
‘an increasing number of studies suggested the use of multigene panel analysis including low, moderate, and high penetrance genes’ (Lines 90-91)
A7: We added two citations (pag 2 lines 56-58, pag 2 lines 90-91).
Q8: It is hard to believe that ‘This research received no external funding’. Perhaps the authors received no specific grant for this particular study; still, the recruitment of 250 subjects, acquisition and processing of the corresponding number of samples, NGS, and Sanger verification demand considerable resources that must have had a source.
A8: No external funding has been received for this study. Our University has an agreement with Italian public health system which reimbursed genetic testing for healthy collaterals and patients with cancer for diagnostic purposes.
Round 2
Reviewer 2 Report
Comments and Suggestions for Authors
The authors have provided a list of all of the variants detected. This is not informative. I think this needs to be changed to include the variants detected in only those patients with cancer. Within this table, the authors need to provide the exact age at which the cancer was diagnosed. Providing a breakdown of the age of either >45 or <45 is not informative. Also, by stating <45 and >45 the authors have not included 45. Should it instead be > or = 45 and <45.
Comments on the Quality of English LanguageSee below.